# Insight into the Evolving Role of PCSK9

**DOI:** 10.3390/metabo12030256

**Published:** 2022-03-17

**Authors:** Mateusz Maligłówka, Michał Kosowski, Marcin Hachuła, Marcin Cyrnek, Łukasz Bułdak, Marcin Basiak, Aleksandra Bołdys, Grzegorz Machnik, Rafał Jakub Bułdak, Bogusław Okopień

**Affiliations:** 1Department of Internal Medicine and Clinical Pharmacology, School of Medicine in Katowice, Medical University of Silesia in Katowice, 40-007 Katowice, Poland; m.kosowski12@gmail.com (M.K.); marcin.hachula@gmail.com (M.H.); marcincyrnek91@gmail.com (M.C.); lbuldak@gmail.com (Ł.B.); marcinbasiak@o2.pl (M.B.); aleksandra.boldys@gmail.com (A.B.); gmachnik@sum.edu.pl (G.M.); bokopien@sum.edu.pl (B.O.); 2Institute of Medical Sciences, University of Opole, 45-040 Opole, Poland; rafal.buldak@uni.opole.pl

**Keywords:** PCSK9, cholesterol, anti-PCSK9, cardiovascular diseases, genetic, atherosclerosis, alirocumab, evolocumab, inclisiran

## Abstract

Proprotein convertase subtilisin/kexin type 9 (PCSK9) is the last discovered member of the family of proprotein convertases (PCs), mainly synthetized in hepatic cells. This serine protease plays a pivotal role in the reduction of the number of low-density lipoprotein receptors (LDLRs) on the surface of hepatocytes, which leads to an increase in the level of cholesterol in the blood. This mechanism and the fact that gain of function (GOF) mutations in *PCSK9* are responsible for causing familial hypercholesterolemia whereas loss-of-function (LOF) mutations are associated with hypocholesterolemia, prompted the invention of drugs that block PCSK9 action. The high efficiency of PCSK9 inhibitors (e.g., alirocumab, evolocumab) in decreasing cardiovascular risk, pleiotropic effects of other lipid-lowering drugs (e.g., statins) and the multifunctional character of other proprotein convertases, were the cause for proceeding studies on functions of PCSK9 beyond cholesterol metabolism. In this article, we summarize the current knowledge on the roles that PCSK9 plays in different tissues and perspectives for its clinical use.

## 1. Introduction

Proprotein convertases (PCs) are a family of nine serine proteases, which also includes proprotein convertase subtilisin/kexin type 9 (PCSK9). Each of those proteases plays a key role in post-translational modifications of propeptides leading to the formation of mature particles e.g., growth factors, enzymes, hormones, and transcriptional factors. Taking into consideration an ability for the activation of many substrates, to date, there seem to be a lot of physiological and pathophysiological processes that PCs take part in [1,2,3]. The tissue localizations and main functions of PCs are summarized in the Table 1.

There are five specific sections in the primary chemical structure of PCs:-Signal peptide, responsible for the exit of enzyme from endoplasmatic reticulum (ER);-Prosegment, maintaining the conformation of peptide chain;-Catalytic domain, responsible for binding to the substrate (according to the substrate specificity and structure of catalytic domain, three types of PCs are distinguished: kexin-like, pyrolysin-like, proteinase K-like);-P domain, stabilizing PCs;-C-terminal domain, characteristic for each PC and determining its cellular localization (this domain can be consisted of: cytoplasmatic tail, transmembrane domain, cysteine-rich domain (Cys-rich domain-CRD), cysteine-histidine-rich domain (Cys-His-rich domain-CHRD), undefined region) [1,2,3].

Figure 1 presents the schematical structure of human PCs [1,2,3].

## 2. PCSK9 and Its Inhibitors

PCSK9, originally known as neural apoptosis-regulated convertase 1 (NARC-1), was discovered in 2003 by Seidah et al. [4]. It is synthetized in the endoplasmic reticulum (ER) of hepatic cells in the form of proenzyme, where after autocatalytic activation, it is transferred to plasma. The soluble PCSK9 binds to the epidermal growth factor homology domain A (EGF-A) of the low-density lipoprotein receptors (LDLRs) located mainly on the surface of hepatocytes, and makes them unable to attach low-density lipoprotein cholesterol (LDL-C) particles. In addition, it promotes the degradation of LDLRs by enhancing their endocytosis and blocking their recycling. Recent research shows that also intracellular PCSK9, before its exocytosis, can reduce the number of LDLRs (intracellular pathway) [2,5]. The targets for degradation by PCSK9 are also very-low-density lipoprotein receptors (VLDLRs) as particles from the close family to LDLRs [6]. The above-mentioned mechanisms lead to the impaired transport of cholesterol from the blood into cells and result in hypercholesterolemia. Furthermore, gain of function (GOF) mutations in *PCSK9* genes are the cause of familial hypercholesterolemia with significantly higher levels of cholesterol in the blood [7], whereas loss-of-function mutations (LOF) are associated with hypocholesterolemia [8]. Table 2 presents the most frequent mutations of *PCSK9* that naturally occur [8,9].

Among factors that affect the expression of *PCSK9* we can single out: age, gender, diet, aerobic exercises, pregnancy, diurnal rhythm, diseases of thyroid gland, kidneys and liver, type 2 diabetes mellitus (DM2), obesity, drugs (e.g., statins) [10,11].

The clinical benefit from discoveries concerning PCSK9 influence on lipid metabolism, was the implementation to the therapy of hypercholesterolemia, medications that inhibit function of PCSK9–monoclonal antibodies against circulating *PCSK9* (*PCSK9* mAbs: alirocumab and evolocumab) and inclisiran—a small interfering RNA (siRNA) which prevents from the translation of messenger RNA (mRNA) of *PCSK9* and in that way decreases the production of mature protein in liver [12,13,14].

Monoclonal antibodies against PCSK9 have the ability to lower plasma LDL cholesterol (LDL-C) by 60% from the baseline. They also decrease the level of apolipoprotein B (apoB) by 50% and triglycerides (TGs) by 15%, and have a slight positive effect on plasma high-density lipoprotein cholesterol (HDL-C) level and increase it by 5–10%. In contrast to statin therapy, they reduce (by even 25%) the concentration of lipoprotein(a) (Lp(a)), which is acknowledged as an independent cardiovascular risk factor [15,16]. The lipid-lowering potential of inclisiran is similar to PCSK9 mAbs, especially with regard to LDL-C and Lp(a) levels [17]. A potential advantage of therapy with inclisiran in the maintenance of compliance results from chemical structure and mechanism of action which ensure longer duration of lipid-lowering effects and, thus, less frequent administration of the drug in comparison with PCSK9 mAbs [18,19]. On the other hand, in spite of the high lipid-lowering potential of PCSK9 inhibitors, therapy with those drugs is still limited, mainly because of the high cost, strict reimbursement criteria and the COVID-19 pandemic, which results in a decline in number of patients that achieve therapeutic goals concerning LDL-C concentration [20,21].

Figure 2 shows the cellular mechanism of PCSK9 and its inhibitors on cholesterol metabolism [22].

High efficacy in reducing the cardiovascular risk by PCSK9 mAbs observed in clinical trials (FOURIER, ODYSSEY Outcomes) seems not to be completely dependent on lipid-lowering effects [3]. The expression of PCSK9 in tissues not directly associated with cholesterol metabolism (e.g., endothelium, kidneys, pancreas, central nervous system), pleiotropic effects of other lipid-lowering drugs (e.g., statins) and multifunctional character of other proprotein convertases, were the reasons for proceeding studies on functions of PCSK9 that go beyond cholesterol metabolism [1,2,3,23,24]. In this article, we summarize current knowledge on functioning PCSK9 in different tissues and perspectives for its clinical use. 

## 3. PCSK9 and Atherosclerosis

One of the main causes for the development of atherosclerosis is the deposition of excess LDL-C within the subendothelial matrix of selected arteries. Then transformed in the processes of oxidation, lipolysis and proteolysis into more reactive form—oxidized low-density lipoprotein (ox-LDL), it plays a key role in the initiation of atherogenesis [25]. Studies from the last few years have confirmed that PCSK9 accelerates the development of atherosclerosis due to the mechanism associated with increasing plasma concentration of LDL-C, but also by direct influence on the cells which build the arterial walls and atherosclerotic plaques [26,27,28]. Analysis of the metabolic pathways and PCSK9 functions in the vascular walls allows to predict the potentially beneficial anti-atherosclerotic effects associated with the therapeutical use of PCSK9 inhibitors [29]. 

The main anti-atherosclerotic effect of PCSK9 inhibitors results from their lipid-lowering efficiency. Data from the OSLER study indicate a significant reduction of LDL-C in the group of patients using evolocumab [30]. Similar results were observed in patients included in the ODYSSEY LONG TERM study, whose plasma LDL-C levels, after using alirocumab, were reduced by up to 62% compared with the placebo group [31].

Previous experiments revealed that PCSK9 inhibitors have a positive effect on the stabilization and morphology of atherosclerotic plaques, making them less vulnerable [32,33]. They also improve the function of platelets by reducing their thrombogenic potential [34,35,36]. Studies in which intravascular ultrasound (IVUS) was used suggest that the concentration of PCSK9 affects the size of the necrotic core within the atherosclerotic plaque, regardless of the concentration of LDL-C [26]. More detailed research carried out in 2020 showed that the use of PCSK9 inhibitors in therapy does not affect the size of the entire atherosclerotic plaque, but significantly improves its stabilization by reducing the lipid core burden index [37].

Arterial stiffness is acknowledged as one of the early predictors of cardiovascular disease [38]. For its indirect noninvasive assessment, pulse wave velocity (PWV) is widely used [39]. Positive correlation between circulating PCSK9 levels and arterial stiffness suggests another way (beyond lipid mechanism) for PCSK9 to affect cardiovascular risk [40]. Studies confirm improvements in arterial stiffness during therapy with PCSK9 inhibitors in patients with familial hypercholesterolemia [41,42,43]. 

### 3.1. Inflammation

Inflammatory processes play an important role in the pathophysiology of atherosclerosis [44]. Their intensification is associated with the development of atherogenesis. PCSK9 is capable of inducing the expression of pro-inflammatory cytokines such as tumor necrosis factor α (TNF-α), interleukin-1β (IL-1β) or interleukin-6 (IL-6). Moreover, it enhances the translocations of transcriptional factors for pro-inflammatory cytokine genes into the cell nucleus and reduces the formation of anti-inflammatory cytokines in macrophages [45,46]. Furthermore, it was shown that PCSK9 regulates the concentration of sirtuins—a family of proteins involved in histone deacetylation, playing a key role in metabolic driver inflammation [47,48].

One of the first meta-analyses carried out on the influence of PCSK9 inhibitors on the inflammatory process did not show any correlation between these drugs and the concentration of C-reactive protein (CRP), a basic marker of the ongoing inflammatory process assessed in clinical practice [49]. Nevertheless, in the assessment of any single biomarker (such as CRP), a certain percentage of subjects with false positive and false negative results should be taken into consideration. To avoid these biases and improve diagnostic sensitivity, biomarker panels and index scores have been introduced in research for several years [50]. On the other hand, some studies reveal the beneficial effects of inhibition of PCSK9 using siRNA on lowering the concentrations of pro-inflammatory cytokines such as interleukin-1β (IL-1β), IL-6 and TNF-α [46,51].

Neutrophil-to-lymphocyte ratio (NLR) and monocyte-to-HDL-cholesterol ratio (MHR) are the novel, widely available markers of inflammation in cardiovascular diseases [52]. High NLR or MHR ratio increases cardiovascular risk [43,53]. Studies show that PCSK9 inhibitors may improve the inflammatory status of patients with familial hypercholesterolemia (FH) described with the use of the above-mentioned parameters [43,53].

### 3.2. Monocytes, Macrophages and Foam Cells

The cells responsible for the secretion of PCSK9 in the vessels are smooth muscle cells (SMCs) and the endothelial cells [54]. Fully functional protein is also detected in atherosclerotic macrophages [55].

The process initiating the development of atherosclerotic plaques is the transformation of monocytes and macrophages into foam cells due to the accumulation of ox-LDL inside them. The influx of lipoproteins through the cell membrane takes place with the participation of many proteins, such as scavenger receptors (SRs), CD36, CD68, lectin-like ox-LDL receptor-1 (LOX-1) [56]. Macrophages excrete the excess of toxic cholesterol into the extracellular space and HDL-C by using membrane transporters, e.g., Adenosine Triphosphate Binding Cassette A1 (ABCA1). PCSK9 shifts the balance of cholesterol transport towards the interior of macrophages by regulating the expression of appropriate membrane proteins, contributing to the formation of foam cells and intensification of atherogenesis [45,57,58].

Apolipoprotein E (apoE), produced in macrophages and smooth muscle cells, is an anti-atherosclerotic protein. It acts via apolipoprotein E receptor-2 (apoER2), which reduces intracellular lipoprotein accumulation and inhibits the formation of foam cells and promotes the anti-inflammatory phenotype of macrophages. [59]. PCSK9 reduces apoER2 expression, attenuating the protective effect of apoE [6].

A very important mechanism of action of PCSK9 inhibitors, which reduces the diapedesis of monocytes into atherosclerotic plaques, is associated with the elevation of the concentration of anti-inflammatory interleukin-10 (IL-10). The increase in its concentration leads to the drop of the expression of TNF-α and C-C chemokine receptor type 2 (CCR2), which are responsible for the influx of monocytes into the atherosclerotic plaque [60,61,62].

### 3.3. Endothelial Cells

Endothelial cell apoptosis promoted by ox-LDL increases its dysfunction and creates favorable conditions for the development of atherosclerosis [56]. Experiments on human endothelial cells obtained from umbilical cord blood indicate that PCSK9 is involved in the enhancement of apoptosis caused by ox-LDL via the Bcl/Bax–caspase-9–caspase-3 pathway [63]. 

In response to the disturbed balance between the accumulation and removal of excess ox-LDL from the foam cells of the subendothelial matrix, the endothelial cells that are lining them receive a signal to produce pro-inflammatory and adhesive cytokines. One of the signal components is the presence of PCSK9 [64].

Reactive oxygen species (ROS) produced in excess in mitochondria, e.g., in the course of inflammation, a cornerstone in the pathogenesis of atherosclerosis [44], are capable of inducing endothelial cells damage, as well as activating inflammatory cells and thus intensifying the inflammatory process within the arterial wall [56]. Cells with silenced *PCSK9* genes produce fewer ROS in their mitochondria [54], which may lead to the conclusion that inhibition of PCSK9 might reduce the risk of endothelial damage. To date, there are no clinical trials that would confirm such an effect of PCSK9 inhibitors.

### 3.4. Smooth Muscle Cells (SMCs)

Under the influence of PCSK9, smooth muscle cells acquire the ability to proliferate, migrate, synthesize collagen and uptake lipoproteins [56]. This cumulatively accelerates the formation of atherosclerotic plaques [65]. Moreover, under the influence of PCSK9, within SMCs, there is an increase in the production of vascular cell adhesion molecule 1 (VCAM-1), facilitating the process of macrophage infiltration into the atherosclerotic plaque [66]. So far, no studies have been carried out to assess the effect of PCSK9 inhibition on the concentration of adhesive factors in SMCs.

### 3.5. Coagulation and Platelet Aggregation 

After several clinical trials, such as JUPITER, showed that the use of lipid-lowering therapy with statins reduces the cardiovascular risk much more strongly than it would result just from the decrease in plasma lipids, scientists began to consider other beneficial mechanisms responsible for this phenomenon [67]. Similar observations regarding cardiovascular risk were also made in the case of PCSK9 inhibitors [68]. 

One of the possible explanations indicates that cardiovascular risk might be related to thrombotic processes caused by inflammation in the vascular endothelium. CD46 and LOX-1 are involved in them [69,70] and, accompanied by ox-LDL binding protein, play a key role in the formation of blood clots [71]. A separate mechanism is associated with the toll-like receptor 2 (TLR-2) stimulation, which activates the process of platelet aggregation through lipid-peroxide-modified phospholipids in the transport of Lp(a) [72,73,74].

Due to the mechanisms described above, the use of PCSK9 inhibitors may limit the process of platelet aggregation in several ways, thus reducing the cardiovascular risk. The first one is by lowering the cholesterol level in the cell membrane of platelets, which results in the drop of their activity [75]. The second one is by decline in the LOX-1 and ox-LDL concentration [54,76]. The ultimate is by reduction of Lp(a) plasma level which decreases the activity of platelets via peroxide-modified phospholipids [77]. The above-mentioned ways of weakening the activity of platelets by inhibiting PCSK9 were unequivocally confirmed in a clinical trial from 2017, with the use of alirocumab and evolocumab, and associated with the reduction of cardiovascular risk [75].

Noteworthy is the influence of PCSK9 inhibitors on the incidence of venous thromboembolism, which is related to the inflammatory process in the endothelium and the atherogenesis [78,79]. The studies conducted so far have not shown a correlation between the concentration of LDL-C and the occurrence of venous thromboembolism [80], however, such a correlation was observed when Lp(a) levels were taken into account [81]. It is crucial in case of PCSK9 inhibitors, which in contrast to statins reduce both, to consider the plasma concentration of LDL-C, and Lp(a) [82]. For this reason, clinical trials have been conducted to assess the impact of alirocumab on the incidence of venous thromboembolism. The results clearly confirmed the beneficial effect of alirocumab therapy on the reduction of the risk of venous thromboembolism incidents, which was associated with a significant reduction in Lp(a) concentration [31]. The second plausible antithrombotic mechanism of PCSK9 inhibitor action, which requires further experimental studies, is associated with their ability to increase the clearance of blood-clotting factor VIII (FVIII)—the essential protein in coagulation processes [83].

## 4. PCSK9 and Heart

Observations on animal models confirmed the expression of PCSK9 in terminally differentiated cardiomyocytes [84]. Furthermore, recent studies have revealed the possible associations between heart disease and PCSK9 independently from the cholesterol regulations [85,86].

Increased expression of PCSK9 after acute myocardial infarction (AMI), triggered by hypoxia and ongoing inflammatory processes, leads to the activation of excessive apoptosis in cardiac cells resulting in heart muscle damage [87]. Negative effects of PCSK9 action, including heart remodeling or increase in infarct size, involve low-density lipoprotein receptor-related protein 1 (LRP1)—the target receptor for PCSK9 in ischemic heart [88]. PCSK9 inhibitors in rat models turned out to have a cardioprotective role in reduction of infarct size, apoptosis and arrhythmias or improving cardiomyocytes contractility [89].

Plasma levels of PCSK9 are increased in patients with heart failure (HF) [87]. There is a positive correlation between circulating PCSK9 and all-cause mortality in the course of HF [90]. The role of PCSK9 in pathophysiology of HF is still unclear, but it might concern the regulation of free fatty acids metabolism, increasing levels of ox-LDL or the influence on LRP1-dependent contractility of heart muscle [91,92,93]. 

In subjects with atrial fibrillation (AF), PCSK9 may promote platelet activation and inhibit the clearance of FVIII, which could result in the increase of risk of arterial thrombosis complications [94,95,96]. PCSK9 is also a protein involved in progression of calcific aortic valve disease (CAVD) by controlling Lp(a) metabolism [97]. 

## 5. PCSK9 and Gastrointestinal System

### 5.1. Pancreas

Beyond the downregulation of LDLRs on the surface of hepatocytes, PCSK9, as a protein expressed in different tissues, could also decrease the number of receptors for LDL in extrahepatic cells [98]. On the surface of pancreatic βcells islets, LDLRs are plentifully expressed. In vitro, islet β-cell lines exposed to LDL were at higher risk of necrosis or apoptosis. Moreover, they presented reduced insulin secretion response to glucose level [99]. The positive lipid-lowering effect of transportation LDL particles from blood into liver cells caused by PCSK9 inhibitors, in other organs, e.g., pancreas, could hypothetically provoke dysfunction of their cells. 

Previous experience with statin treatment showed that lipid-lowering treatment may worsen glucose metabolism [100]. During statin therapy, a significant rise in LDL-R in liver and a concomitant rebound in PCSK9 serum concentration are seen [101]. As a result of these observations, a thorough evaluation of relationship between PCSK9 and glucose metabolism was performed. Human observational studies revealed that loss-of-function mutations of PCSK9 are linked to elevated fasting plasma glucose and increased risk of DM2 [102]. However, compared to healthy subjects, PCSK9 levels were significantly higher in those with impaired fasting glycaemia (IFG), impaired glucose tolerance (IGT) and DM2 [103,104]. A mice model (PCSK9 “knock out” mice) showed impaired insulin secretion resulting from cholesterol accumulation in islet beta cells [105]. However, these finding may be different in case of a therapeutic approach. Initial results of the DESCARTES study showed that evolocumab was devoid of the influence of glucose metabolism in diabetes-prone populations (impaired fasting glycemia, metabolic syndrome) and subjects without dysglycemia [106]. Recent data from meta-analysis of RCTs on the effects of various PCSK9 inhibitors (evolocumab, alirocumab and inclisiran) have not shown increased incidence of new-onset diabetes (RR 0.99; 95%CI: 0.86–1.13) [107]. Similar data come from “real-world” pharmacovigilance studies [108]. PCSK9 inhibitors are not associated with increased incidence of new onset diabetes, but a larger number of hyperglycemic adverse events has been observed (predominantly in patients with established diabetes). These discrepancies in the influence on glucose metabolism between therapeutic use and both loss-of-function and KO models may derive from different mechanisms. Monoclonal antibodies block circulating PCSK9, whereas KO mice and subjects with loss-of-function mutations are devoid of synthesis of PCSK9 in all tissues. It was noted that local *PCSK9* expression may be involved in the function of insulin-secreting cells [109]. 

Diabetic dyslipidemia observed in patients with DM2 elevates cardiovascular risk [110]. It is characterized by an increase in TGs, LDL, small-dense LDL (sdLDL) and by a decrease in HDL and apolipoprotein A1 [111]. Due to the high efficacy of PCSK9 inhibitors and the wide influence on lipid abnormalities that play a key role in diabetic dyslipidemia, these drugs are indicated for patients at very high cardiovascular risk when conventional therapy with statins and ezetimibe are insufficient [112].

### 5.2. Small Intestine

A small intestine is the second major organ involved in cholesterol metabolism. It is responsible for lipid absorption and elimination, as well as the synthesis of HDL fraction. [113]. Intestinal cholesterol absorption, inhibited by ezetimibe, is used in patients who have failed to achieve target levels of plasma LDL with statin therapy. It takes place especially in epithelial cells of small intestine by blocking the Niemann–Pick C1-like 1 (NPC1L1) protein [114]. According to the current knowledge, PCSK9 produced in enterocytes is not secreted into the systemic circulation. Its action is related to auto- and paracrine effects and leads to increasing the clearance of triglyceride-rich lipoproteins via LDLR and reducing apolipoprotein B48 levels. This results in the decrease of postprandial triacylglycerol (TG) levels. [115] Nowadays, it is widely known that PCSK9 is one of the regulators of intestinal triglycerides metabolism [116].

### 5.3. Liver

Non-alcoholic fatty liver disease (NAFLD) is a progressive disorder caused by the abnormal accumulation of triglycerides within hepatocytes. Hepatic steatosis could be recognized when more than 5% of liver cells contain droplets of TGs. NAFLD could progress to non-alcoholic steatohepatitis (NASH), characterized by inflammation, and finally to cirrhosis, which is the main risk factor of hepatocellular carcinoma (HCC) [117]. NAFLD is highly associated with the increased risk of cardiovascular disease (CVD) and DM2 [118]. The participation of PCSK9 in the metabolism of TGs at the intestinal level suggests the association between PCSK9 plasma concentration and NAFLD pathogenesis [115,119].

A study of 201 patients showed a positive correlation between circulating PCSK9 and hepatic steatosis grade. It also revealed that PCSK9 deficiency resulted in resistance to liver steatosis [120]. Furthermore, in another study with 698 participants, it was demonstrated that circulating PCSK9 levels were firmly related to all plasma biomarkers of liver function and the presence of hepatic steatosis. [117] On the other hand, a study of 478 subjects with DM2 or metabolic syndrome showed that plasma level of PCSK9 was not associated with liver steatosis [121].

The above-mentioned data suggest that a high PCSK9 level could play a key role in liver lipid storage and metabolism of TG, thus contributing to the pathogenesis of NAFLD. Thereby, PCSK9 inhibitors could be a potential therapeutic option for patients with NAFLD.

In the study conducted by Scicali R. et al., PCSK9 inhibitors significantly ameliorated steatosis biomarkers in patients with FH and low TG/HDL ratio. [122]. Additionally, a retrospective observation confirmed a positive effect of PCSK9 inhibitors on liver function in NAFLD. [123] A case report of a patient with NASH showed that therapy with evolocumab resulted in a more-than-80% reduction of liver transaminases. A control liver biopsy revealed a normalization of liver histology [124]. In a study of alcohol liver disease (ALD) in animals, treatment with alirocumab significantly decreased the PCSK9 level, which resulted in weakening of fatty infiltration, inflammation, oxidative stress, and hepatic cell damage [125]. This shows that inhibitors PCSK9 could be useful in liver diseases such as steatosis. Large randomized clinical trials are necessary to confirm this hypothesis.

## 6. PCSK9 and Kidneys

Kidneys are organs able to synthetize PCSK9, which was confirmed by studies in humans and animals [126]. The number of LDLRs on the surface of renal cells depends on PCSK9 [98]. High levels of PCSK9 can provoke excessive accumulation of lipids in kidneys and lead to their fibrosis. By using the vaccine against PCSK9, these pathological processes could be diminished [127].

The discovery that hypercholesterolemia in patients with nephrotic syndrome is associated with elevated levels of PCSK9 enabled the successful use of evolocumab in this novel indication [128,129,130]. Further studies are necessary to assess the safety and effectiveness of this treatment.

*PCSK9* is also involved in the expression of renal epithelial sodium channel (ENaC) responsible for the regulation of blood pressure by controlling sodium reabsorption from terminal part of the nephron. Nevertheless, to date, observations in mice with deficiency in PCSK9 have not confirmed any changes in sodium concentration or in blood pressure [131,132]. 

The main cause of death in patients with chronic kidney disease (CKD) are cardiovascular complications. They partially result from CKD-associated dyslipidemia characterized by elevated levels of Lp(a), TGs, VLDL and low levels of HDL-C [133]. The data concerning PCSK9 concentration in patients with CKD are inconsistent [134]. Taking under consideration the high efficacy of PCSK9 inhibitors in decreasing cardiovascular risk in a lipid-lowering and probably a beyond lipid-lowering way, they have potential to be useful in that group of patients [5].

## 7. PCSK9 and the Endocrine System

Data on the impact of PCSK9 on endocrine systems are scarce. The majority of available manuscripts describs simple associations between concentration of PCSK9 and selected endocrine disorders. The causal relationship in many cases is still missing, but several findings seem promising and depict PCSK9 as a potential novel marker of diseases. 

### 7.1. Thyroid Function

Thyroid function has an important impact on lipid profile. Hypothyroidism is one of the common causes of secondary hyperlipidemias [135]. Elevated TSH level leads to increased expression of *PCSK9* [136]. On the other hand, hyperthyroid patients tend to have lower cholesterol levels, which is associated with reduction in PCSK9 serum concentration [137]. Little is known about the potential impact of PCSK9 on thyroid function. No significant correlations between the level of PCSK9 and autoantibodies against thyroperoxidase and thyreoglobulin have been observed [138]. However, elevated levels of PCSK9 have been found in patients with Graves’ orbitopathy and its concentration has positively correlated with autoantibodies against TSH receptor and clinical activity score. In vitro studies showed that PCSK9 inhibition significantly reduced pro-inflammatory cytokines production, oxidative stress markers and adipocytes formation in Graves’ orbitopathy [139]. Those observations suggest that PCSK9 level might be considered as a potential marker in the course of Graves’ orbitopathy.

### 7.2. Sex Hormones

PCSK9 level varies among gender, but the data on the subject yield conflicting results. Some studies indicate that it is higher in women than in men and postmenopausal women [140]. Others have suggested that PCSK9 level is higher in postmenopausal than in premenopausal women [141]. PCSK9 also tends to rise with increasing age. A connection between sex hormones and PCSK9 was supported by data on elevated PCSK9 during pregnancy—a physiological condition with increased levels of sex hormones [142]. Several studies showed direct association between the PCSK9 level and pharmacological interventions. In an animal model, a 45% reduction in PCSK9 during high-dose estrogen therapy was noted [143]. However, data from human interventional studies indicated that hormone replacement therapy in postmenopausal women and in men during hormonal ablation therapy with gonadotropin-releasing hormone (GnRH) agonist combined with androgen receptor inhibitor has not shown any statistically significant impact on PCSK9 level [141]. Data from clinical trials (DESCARTES) showed lack of impact of evolocumab on estradiol and testosterone level during a 52-week treatment. Interestingly, during the above-mentioned study, FSH (6.3 IU/l vs. 12.0 IU/l; *p* < 0.05) and LH (7.6 IU/l vs. 10.8 IU/l) in women increased, while the impact on FSH and LH in males was negligible [144]. The causal relationship between sex hormones and PCSK9 levels warrants further exploration. The discrepancy between human and animal studies may result from supraphysiological doses of hormones in the latter experiments.

Functioning of the reproductive system is partially dependent on the cardiovascular system. Erectile dysfunction is common in patients with atherosclerosis, hypertension, DM2 and metabolic syndrome [145]. PCSK9 mAbs was found to be effective in improvement of sexual dysfunction in male with familial hypercholesterolemia [41].

### 7.3. Adrenals

High efficacy of PCSK9 inhibitors in lowering plasma cholesterol level was a cause for concern in terms of disturbances in adrenal steroidogenesis in which cholesterol plays a basic role [146]. In patients with loss-of-function (LOF) mutations in *PCSK9*, no adrenal dysfunction was observed [147]. The basal cortisol production during the therapy with evolocumab was preserved as well [144]. Nevertheless, recent studies suggest that adrenal stress response during the therapy with PCSK9 inhibitors can be reduced [148]. Clinical importance of the above-mentioned observations and the missing data that link adrenal stress response and PCSK9 function require further exploration.

Elevated levels of aldosterone are correlated with mortality and the occurrence of acute ischemic events [149]. The therapy with evolocumab decreases baseline and stimulated levels of aldosterone, which could positively influence cardiovascular risk [150]. Further studies are needed to confirm that hypothesis. 

### 7.4. Polycystic Ovary Syndrome (PCOS)

PCOS is one of the most common endocrinopathies affecting 5–10% women with child-bearing potential [135]. The precise pathophysiology remains unknown, but several features are seen in majority of cases (i.e., elevated androgens production, increased insulin resistance, lipid profile abnormalities). Commonly, patients with PCOS suffer from atherogenic dyslipidemia [135], which is characterized by elevated levels of atherogenic, sdLDL-C particles. Cholesterol is a substrate for the production of steroid hormones. One of the sources of intracellular cholesterol is the uptake of LDL particles by thecal and granulosa cells via LDL-R. The density of LDL-R is dependent on the level of PCSK9, which has been found to be significantly higher in women with PCOS [151]. In a mice model of PCOS, it was also noted that PCSK9 expression is significantly increased in the ovaries, which led to a decreased uptake of cholesterol due to diminished density of LDL-receptors [152]. In previous studies, it was observed that low levels of LDL-R may lead to subfertility [153]. Treatment with alirocumab caused an increase in the expression of LDL-R, which was connected with improved uptake of cholesterol in ovarian cells. This phenomenon might indicate a potential connection between PCSK9 and PCOS. PCSK9 level was positively correlated with androgen and LH levels and negatively correlated with FSH levels in women with PCOS [153]. Whether the connection is based on the LDLR level, lipid profile alteration or direct impact of PCSK9 still remains not completely understood. Currently, PCSK9 polymorphisms are studied as a potential factor in the development of PCOS [154].

## 8. PCSK9 and Central Nervous System (CNS)

Since cholesterol is the basic component of neuronal cell membranes, the question of whether the use of intensive lipid-lowering therapy may adversely affect their functioning has circulated for years. In particular, it concerns the type of therapies that affect the concentration of PCSK9. 

Research has shown that PCSK9 is involved in the development of the central nervous system (CNS) and the process of proper closure of the neural tube in embryos [4,155]. On the other hand, the analysis of the population included in the PROSPER study revealed no relationship between the loss-of-function mutation in *PCSK9* and neurocognitive disorders [156].

Observations from studies suggesting worse results in neurocognitive tests and a lower psychomotor processing speed in patients treated with statins [157], better scores achieved in neuropsychological tests in a group of healthy adults with higher level of LDL [158] or increased concentration of PCSK9 in the cerebrospinal fluid in patients suffering from Alzheimer’s disease (AD) [159] were the causes of analyzing the potential adverse effects of PCSK9 inhibitors on the CNS. Although theoretical models suggest that exposure to PCSK9 inhibitors may increase the risk of AD [160], it has not been confirmed by the experimental data so far. [161]. Additionally, meta-analyses and reviews of numerous randomized clinical trials did not reveal higher frequency in neurological disorders (including neurocognitive ones) with the use of PCSK9 inhibitors compared to the control groups [162,163,164,165].

One of the most common causes of cognitive disorders is vascular dementia—a disease directly related to the process of atherogenesis within the arterial vessels of CNS [166]. The prevention of adverse cardiovascular events, such as ischemic strokes, is the main goal of lipid-lowering therapy. Published in May 2020, the results of a randomized clinical trial using evolocumab added to prior statin therapy revealed the effectiveness of PCSK9 inhibitors in both primary and secondary prevention of ischemic stroke [167]. Furthermore, meta-analysis of 16 clinical trials showed a significant reduction in the risk of ischemic stroke in patients treated with evolocumab or alirocumab. Moreover, it did not confirm any negative influence of these drugs on cognitive functions [168]. In the future, these observations may modify the therapeutic recommendations for patients at high cardiovascular risk.

## 9. PCSK9 and Cancer

The cholesterol level within cancer cells is usually high, but the significance of this finding is still controversial [169]. Tumor cells show a high expression of LDLRs and hydroxymethylglutaryl–coenzyme A (HMG-CoA) reductase, which results in the elevation of intracellular cholesterol and its increased catabolism [170,171]. High cholesterol levels in neoplastic tissues were confirmed in clinical trials concerning mainly breast, prostate and liver malignancies [172,173,174]. Cholesterol accumulation in malignant tissues is considered as their characteristic feature [175].

Some studies suggest a positive correlation between elevated plasma cholesterol level and the initiation of neoplastic processes, but others oppose such an association between cholesterol and carcinogenesis [169]. Other observations indicate the role of cholesterol in the progression of cancer disease [175].

Cholesterol is also an essential component of lipid rafts (cell membrane domains), which contain a large number of neoplastic adhesion and signaling molecules responsible in cancer cells for the migratory and invasion processes [176,177]. CD44 is the major receptor found in lipid rafts and involved in the spreading of cancer cells [178]. Low cholesterol supply causes increased CD44 shedding and results in the suppression of tumor cell migration [176,177,178]. 

PCSK9 plays a key role in cholesterol metabolism by limiting the number of LDLR [179]. Additionally, overexpression of PCSK9 downregulates genes responsible for apoptosis and the immune system response [180,181].

The influence of PCSK9 on the proliferation, metabolism and especially apoptosis of neoplastic cells has been studied by many scientists. The expression of *PCSK9* (originally called NARC-1) was discovered in cerebellar granule neurons (CGN) and induced after damage leading to apoptosis. The studies suggest that NARC-1 may promote apoptotic signaling in brain cells by interacting with mediators such as caspase-3 and death receptor 6 (DR6) [182,183].

Research on the effect of PCSK9 deficiency on melanoma metastases to the liver carried out in mice concludes that the reduced level of PCSK9 elevates the concentration of proapoptotic TNF-α and decreases the level of anti-apoptotic protein bcl-2, which was clinically manifested by the limitation in the number of liver metastases compared to wild-type mice. [184] Furthermore, based on the microarray study, it was found that *PCSK9* overexpression reduced the number of apoptotic genes in cells [185]. In vitro observations of neuroglioma cells revealed *PCSK9* overexpression led to the inhibition of c-caspase expression and stimulation of X-linked inhibitor of apoptosis (XIAP) and phosphorylated form of Akt, also known as protein kinase B (p-Akt/PKB), and consequently caused accelerated growth and development of cancer cells, while *PCSK9* siRNA inhibited neuroglioma cell proliferation [185].

Studies on the effect of *PCSK9* siRNA on lung adenocarcinoma showed that PCSK9 had a pro-apoptotic effect on tumor cells by activating caspase-3 and weakening the functions of XIAP and survivin (a protein that is involved in the division and inhibition of apoptosis). Disturbances in the mitochondrial membrane related to the imbalance between bcl-2/bax and the subsequent release of cytochrome C were also observed. Moreover, in cells transfected with PCSK9, siRNA induction of endoplasmic reticulum stress (ERS) was noticeable. These processes led to cell death [186].

Based on microarray analyzes of hepatocellular carcinoma (HCC) cells, decreased expression of PCSK9 was found. This resulted in reduced internalization of LDLRs and, consequently, increased cholesterol metabolism in neoplastic cells and stimulation of disease progression [187]. On the other hand, the results of other studies showed a correlation between high PCSK9 expression and HCC progression with vascular infiltration and large tumor size. PCSK9 inhibited apoptosis of tumor cells via bax/bcl-2/caspase-9/caspase-3, which consequently stimulated the proliferation of neoplastic tissue [188].

Studies on gastric adenocarcinoma cells show that PCKS9 is overexpressed in up to 60% of cases [189]. Therefore, scientists speculate that PCSK9 may in the future become a valuable biomarker of cancer development [187,189].

The above-mentioned data reveal that PCSK9 plays an important role in the development, metabolism and progression of neoplasms. Not all mechanisms of PCSK9 action on malignant tissues have been fully elucidated yet. Some show conflicting results. Further research is needed to clearly understand the impact of PCSK9 on the oncogenesis processes.

## 10. PCSK9 and Infections

Taking into consideration the fact that LDLRs are involved in the pathogenesis of some infections, the influence of lipid-lowering drugs on the course of infectious diseases, including their life-threatening complications, is interesting to investigate [190].

### 10.1. Bacterial Infections

Lipid molecules which form the structure of bacterial cell membranes, e.g., lipopolysaccharides (LPS) and lipoteichoic acid, play the leading role in initiating the host immune response against pathogens [191]. Studies show that the response intensity is significantly related to the concentration of those lipids, and limiting it prevents the occurrence of septic shock. The clearing of bacterial lipid structures from the bloodstream occurs with the participation of LDLRs [190,191,192].

Results from the experimental sepsis model showed that overexpression in *PCSK9* was characterized by greater bacterial dissemination and consequent organ pathology, particularly in the liver and lungs [193]. Others have proven a significant reduction in proinflammatory cytokines concentration after LPS exposure in the group of patients with *PCSK9* LOF mutations [194]. The above-mentioned data lead to a hypothesis that using PCSK9 inhibitors may improve the prognosis of patients with severe bacterial infections. Currently, this assumption is being verified in a randomized placebo-controlled double-blind study with alirocumab (NCT03634293). The study will assess the difference in survival after 28 days in patients with sepsis or septic shock, treated with PCSK9i or placebo in an intensive care unit (ICU). The results might introduce a novel group of drugs to the therapy of severe bacterial infections [195].

### 10.2. Viral Infections:

Flu-like symptoms and an increased number of upper respiratory tract infections have been reported as side effects in clinical trials with PCSK9 inhibitors [196,197]. So far, it has not been possible to explicitly confirm or contradict that PCSK9 is directly involved in the pathogenesis of viral diseases [198]. Considering the fact that the pathogenesis of viral infections involves many host factors, e.g., lipids and lipoproteins [199], the influence of PCSK9 inhibitors on the course of viral infections has captured the attention of many scientists all over the world.

#### 10.2.1. Hepatitis C Virus

Hepatitis C virus (HCV) infection is a crucial cause of severe liver disease leading to chronic hepatitis, cirrhosis and hepatocellular carcinoma. The World Health Organization estimates that seventy-one million people have HCV infection worldwide and at least 400,000 people die of it every year [200].

HCV enters hepatocytes with the use of different proteins—one of them is LDLR [201]. Syed GH et al. observed that hepatic cells infected by HCV contain more LDLRs than those from the control group [202]. Some studies suggested that LDLR also plays a key role for the post-entry viral processes such as replication [203]. Furthermore, the report from an in vitro study revealed that LDLR degradation by PCSK9 with the participation of hepatic CD81 particle is one of the main ways in which HCV invades cells [204,205]. 

These results raised the concern that PCSK9 inhibitors might increase CD81 levels, and thereby susceptibility to HCV infection. In 2016, Ramanathan A. et al., in their in vitro and in vivo studies, did not confirm that treatment with alirocumab increased the risk of HCV infection or CD81 levels [206]. Moreover, analyses of the results of ten clinical trials, which compared the action of alirocumab with placebo or ezetimibe, showed no elevations in predisposition to HCV infection in the groups of participants treated with PCSK9i. [206] Additionally, the ODYSSEY and FOURIER clinical trials did not reveal statistically significant increases in the incidence of hepatic disorders [207,208].

#### 10.2.2. Dengue Virus

Dengue fever is one of major global health problems, with 390 million individuals infected annually, and with a rate of mortality that may reach up to 20% [209,210]. Dengue virus (DENV) requires cholesterol in the development cycle: viral entry by membrane, fusion, and replication [211]. In vitro and animal models with statins revealed the reduction in DENV infections [212,213]. Unfortunately, lovastatin used in a clinical trial did not repeat this effectiveness in human [214]. New hopes appeared with the efficacy of lipid-lowering drugs from the group of PCSK9 inhibitors. Gan ES et al. reported that elevated PCSK9 levels are associated with higher viremia and an increased risk of a more severe course of dengue [215]. Clinical trials with PCSK9 inhibitors are required to test their potential effectiveness in a novel indication.

#### 10.2.3. SARS-CoV-2

An interstitial pneumonia caused by severe acute respiratory syndrome coronavirus 2 (SARS-CoV-2) was first reported in Wuhan, Hubei, China, in December 2019. As of 14 January 2022, over 318 million SARS-CoV-2 infections have been reported. So far, 5.5 million deaths have been associated with coronavirus disease 2019 (COVID-19). Cardiovascular diseases, arterial hypertension, hyperlipidemia, diabetes, kidney and chronic lung diseases, cancer and obesity have all been associated with severe course of the disease and increased mortality rate [216,217]. 

SARS-CoV-2 enters host cells using angiotensin converting enzyme 2 (ACE2) receptors and with the participation of cholesterol in trafficking ACE2 to the viral entry site [218]. There are some hypotheses suggesting that PCSK9 inhibitors, as the lipid-lowering drugs, could affect various elements in the pathophysiology of SARS-CoV-2 infection [218]. 

COVID-19 is associated with increased incidence of coagulation abnormalities and higher risk of venous thromboembolic disease [219]. An analysis of clinical trials (FOURIER and ODYSSEY OUTCOMES) reported lower rates of venous thromboembolism in patients who were treated with PCSK9 inhibitors compared with placebo [207,208].

Systemic endothelial inflammation is considered as one of the main causes of severe course of SARS-CoV-2 infection [220]. The positive correlation between LDL plasma concentration and endothelial dysfunction is widely known [221]. Some studies have confirmed that PCSK9 inhibitors improves the functioning of the endothelium in patients with inflammatory diseases [222,223]. In the ODYSSEY FH I and FH study, PCSK9 inhibitors decreased the level of the atherothrombogenic Lp(a) by approximately 30% [224]. The observation by Huang W et al. showed a negative correlation between PCSK9 expression and the risk of infection with SARS-CoV2 [225]. Further studies are required to unequivocally assess the potentially beneficial effects of PCSK9 inhibitors in the therapy for COVID-19.

### 10.3. Parasites

Parasites the organisms that need cholesterol to develop properly. As they lack the ability to synthesize it de novo, they use cholesterol from the host [226]. 

Arama C et al. examined the influence of GOF mutations in *PCSK9* on the course of malaria. The study revealed that in a group of 752 Malians children, those who were GOF mutation carriers were prone to a more severe course of disease [227]. In the research of Fedoryak O et al., LOF mutations in *PCSK9* were associated with reduced mortality in the course of malaria [228]. According to their results, a hypothesis that PCSK9 inhibitors could be useful for the therapy and prevention of malaria was formed. To date, there is no scientific evidence confirming the association between the therapy with PCSK9 inhibitors and the course of malaria. Further studies are required. 

## 11. New Approaches to PCSK9 Inhibition

Since drugs currently used in lipid-lowering therapy and affecting the action of PCSK9 have presented high efficacy in cardiovascular risk reduction, novel methods for inhibiting PCSK9 have being developed [229]. 

The first promising group of substances consists of peptide inhibitors of PCSK9. Some of them mimic the EGF-A binding domain of LDL-R and as competitive inhibitors preventing the interaction between LDL-R and PCSK9 [230]. The other group resembles the COOH-terminal domain of PCSK9 and reduces the degradation of LDL-R. [231] Adnectins—small proteins based on the structure of human fibronectin, represent a different type of peptides that show high affinity and specificity for blocking the LDL-R binding domain of PCSK9 [232]. 

The second group of potentially effective PCSK9 inhibitors interacts with nucleic acids connected with PCSK9. Anti-sense oligonucleotides (ASO) lower the levels of PCSK9 mRNA by favoring its degradation by ribonuclease H (RNAse H) [233]. A genome editing technique with clustered regularly interspaced short palindromic repeats (CRISPR) appears to be an innovative approach to PCSK9 inhibition. The CRISPR system is an adaptive form of immunity observed in bacterial cells, which allows to destroy viruses by recognizing their genome on the basis of viral DNA retained in microbial genome during previous infections [234]. CRISPR edited for targeting PCSK9 could ensure long-lasting or even lifelong LDL-C reduction. Promising findings in preclinical models require further study before this new type of therapy could be safely introduced in humans [235]. 

The last group of substances that interfere with PCSK9 are vaccines. By stimulating the immune system, they provoke formation of PCSK9 antibodies with high affinity and long-lasting existence [236]. Studies on mice confirmed their efficacy in lowering the concentration of LDL-C and found positive effects in attenuating atherosclerotic plaques without serious side effects [237,238]. 

## 12. Conclusions

Knowledge about the role of PCSK9 in different tissues has been accumulating since its discovery in 2003. Therapeutic intervention in the PCSK9 pathway has resulted in successful implementation of novel group of drugs in lipid-lowering therapy, which dramatically improved the prognosis of patients at very-high cardiovascular risk. Since then, novel aspects of PCSK9 functioning in selected tissues, e.g., arterial vessels, blood, gastrointestinal tract, CNS, and kidneys have been described. Furthermore, it appears that PCSK9 may play an important role in the initiation of other morbidities. The observation that PCSK9 is involved in the pathophysiology of inflammation, heart disease, atherosclerosis, nephrotic syndrome, hepatic steatosis, infections, and cancer metastases might positively influence on the therapeutic approach to those diseases in the future. Nevertheless, further studies are necessary to fully understand the functioning of PCSK9 in human cells before new therapies interfering with PCSK9 can be introduced. 

## Figures and Tables

**Figure 1 metabolites-12-00256-f001:**
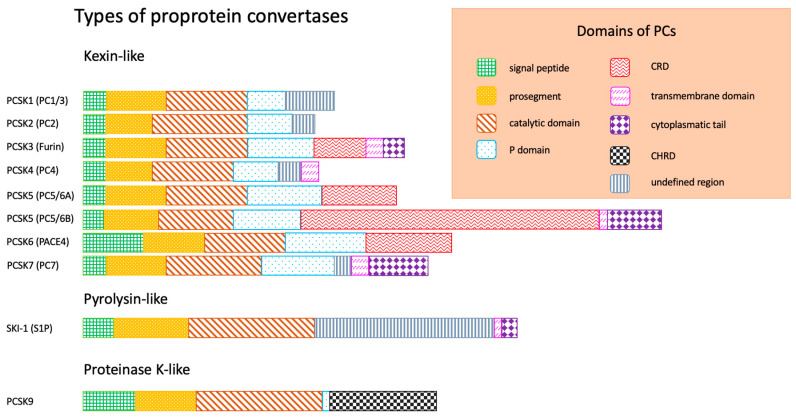
Schematical structure of human PCs [1,2,3]. Abbreviations: CHRD, Cys-His-rich domain; CRD, Cys-rich domain; PCSK1 (PC1/3), proprotein convertase subtilisin/kexin type 1 (proprotein convertase 1/3); PCSK2 (PC2), proprotein convertase subtilisin/kexin type 2 (proprotein convertase 2); PCSK3, proprotein convertase subtilisin/kexin type 3; PCSK4 (PC4), proprotein convertase subtilisin/kexin type 4 (proprotein convertase 4); PCSK5 (PC5/6), proprotein convertase subtilisin/kexin type 5 (proprotein convertase 5/6); PCSK6 (PACE4), proprotein convertase subtilisin/kexin type 6 (paired basic amino acid cleaving enzyme 4); PCSK7 (PC7), proprotein convertase subtilisin/kexin type 7 (proprotein convertase 7); SKI-1 (S1P), subtilisin/kexin isozyme 1 (sphingosine 1-phosphate); PCSK9, proprotein convertase subtilisin/kexin type 9.

**Figure 2 metabolites-12-00256-f002:**
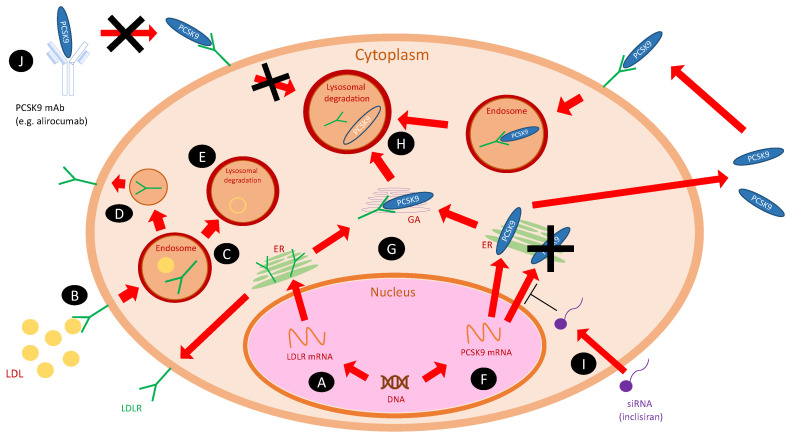
The cellular actions of PCSK9 and its inhibitors on cholesterol metabolism [22]. Abbreviations: ER, endoplasmatic reticulum; GA, Golgi apparatus; LDL, low-density lipoprotein; LDLR, low-density lipoprotein receptor; PCSK9, proprotein convertase subtilisin/kexin type 9; PCSK9 mAb, monoclonal antibody against PCSK9; siRNA, small interfering RNA (inclisiran). (**A**) *LDLR* gene is transcribed into mRNA, then translated within ER into mature protein and transferred on the cell surface. (**B**) LDL particles, after binding to the LDLRs on the cell surface, are internalized into an endosome. (**C**) As the endosome matures, LDL and LDLR decouple—LDLR (**D**) recycles, and LDL is catabolized in the lysosome (**E)**. (**F**) *PCSK9* gene is transcribed into mRNA and then translated within ER into the mature protein. It can bind to the LDLR intracellularly within Golgi apparatus (**G**) or be secreted to the extracellular space—both ways lead to lysosomal degradation of LDLR (**H**). (**I**) Inclisiran (siRNA) inhibits the translation of PCSK9 mRNA and prevents the formation of mature protein. (**J**) PCSK9 mAbs (e.g., alirocumab) bind to the soluble PCSK9 in the plasma and prevent from lysosomal degradation of LDLRs.

**Table 1 metabolites-12-00256-t001:** Basic functions and tissue distribution of proprotein convertases (PCs) [1,2,3].

Proprotein Convertase (PC)	Tissue Distribution	Function
PCSK1 (PC1/3)	Hypothalamus, pituitary, pancreas, thyroid gland, adrenal glands	Regulation of appetite; activation of insulin, glucagone, orexin, ghrelin
PCSK2 (PC2)	Central nervous system (CNS), pituitary, pancreas	Regulation of carbohydrate metabolism
PCSK3 (furin)	Ubiquitous	Regulation of embryogenesis, activation of growth factors and bacterial toxins (e.g., *Clostridioides* spp.), carcinogenesis
PCSK4 (PC4)	Germinal	Regulation of reproduction processes
PCSK5 (PC5/6)	Ubiquitous	Regulation of embryogenesis (e.g., CNS)
PCSK6 (PACE4)		
PCSK7 (PC7)	Colon, spleen, liver	Regulation of lipid metabolism, atherogenesis
SKI-1 (S1P)	Ubiquitous	
PCSK9	Liver, small intestine	

**Table 2 metabolites-12-00256-t002:** Most frequent naturally occurring *PCSK9* mutations [8,9].

*PCSK9* Variants	
Gain of Function (GOF)	Loss of Function (LOF)
E32K, L108R, S127R, D129G, D129N, R218S, D374Y	R46L, C679X, Y142X

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
