# Peer review of "Insight into the Evolving Role of PCSK9"

_metabolites, 2022, doi:10.3390/metabo12030256_

Round 1

Reviewer 1 Report

In this paper, the authors focused on the role of PCSK9 and its inhibition on cholesterol metabolism and beyond. The paper is interesting, the topic has an important scientific impact and the english style is adequate. However, the paper is missing of important studies in this fiels.

Concerning PCSK9 and atherosclerosis, please consider this article (10.1016/j.jacl.2020.01.015) and comment this in the related paragraph.

Concerning inflammation, please consider these articles (10.1016/j.numecd.2021.08.034; 10.1007/s00592-021-01703-z) and comment these in the paragraph.

Concerning platelet activity, please consider this article (10.3390/ijms22137193) and comment this in the paragraph.

I'm very satisfied to see a paragraph concerning the effect of PCSK9 on liver function.

Concerning PCSK9 and atherosclerosis, I also suggest to insert the effect of PCSK9 inhibitors on sexual function, since that atherosclerosis aggravates sexual function; please consider this manuscript (10.3390/jcm9113597) and comment this in the paragraph.

In addition, there are the following three references should be cited to improve the quality of the article.

10.1161/JAHA.117.005764;10.1016/j.arteri.2015.11.001; 10.1016/j.atherosclerosis.2019.07.015

Reviewer 2 Report

I have read in detail the review by Maliglówka et al on the role of PCSK9. I sincerely think that this review does not provide much information on the topic and is not very precise in aspects such as the molecular mechanisms of the regulation of PCSK9 expression. In addition, the same authors published a review on PCSK9 less than a year ago in Int J Mol Sci (reference #30).

This reviewer is not a native English speaker but it seems to me that from a grammatical point of view it should be improved and/or be revised.

In this review, the authors intend to cover broader aspects of the role of PCSK9, although without going into details or specific mechanisms, which makes it useless to the reader who has to search the references to find more precise details.

It has also been difficult for me to find any of the bibliographic citations since they are not in PubMed and they are editorial letters that I think should not be cited for reasons of scientific rigor (for example reference #103).

However, what this reviewer found totally unacceptable is the poor art work in the scheme they present as Figure 2. This scheme is very poor from a design point of view and there are also aspects that cannot be understood and that are essentially incorrect. For example, the point (G) where PSCK9 appears to be bound to the LDL-R in the cytoplasm. In addition, the scheme is very similar to the one published in ref#30.

From a formal point of view, it begins with a very brief Introduction, basically referring to Table 1 and Figure 1. It then goes on to section 2: PCSK9 and its inhibitors. Apart from the 2 monoclonal antibodies and the siRNA that are described, this section could also include the vaccines that have been produced against PSCK9, which are mentioned scattered throughout the review, and perhaps also include a table or a list with the known mutations of PCSK9, which are also mentioned but scattered among the different sections.

I think the authors have made a bibliographical effort to search for and put together the role of PCSK9 in different organs and that most of the references are recent, which may allow the reader to find details that the review does not provide. But, I also think that some of the bibliographic citations are irrelevant because they are published in journals that are difficult to access (low visibility and dubious credibility).

Finally, a recent extensive and detailed review of PCSK9 in chronic liver disease has been published in Int J Mol Sci (2022); this is a good example of a review containing useful concepts and data and in fact, the overlap with the present review in many aspects. 

Reviewer 3 Report

Dear Editor,

I carefully read the manuscript by Maligłówka et al. that is interesting and potentially of great interest for the readers of the Journal. As a matter of fact, the article touchs upon hot topics (especially after inclisiran has been approved by FDA).

My comments and suggestions for the authors are the following:

  • Parts of the manuscript are not formatted following the Instructions for Authors of the Journal
  • All abbreviations used in the figures should be defined at the bottom of the figures
  • English language needs to be carefully revised (especially as concerns verbal concordances)
  • The authors should consider to refer to PMID: 28468788, PMID: 34596005, PMID: 35159928 and PMID: 35112773  in their manuscript.

Round 2

Reviewer 1 Report

The authors adequately satisfied the requested revision.

Reviewer 2 Report

Dear Dr Okopién,

I think you did a good revision and now the manuscript is acceptable for publication in Metabolites.

All the best.

Reviewer 3 Report

Dear Editor,

I carefully read the revised version of the manuscript that is significantly improved in comparison with the previous version. I recommend its publication in the Journal.